# Lysine-Dendrimer, a New Non-Aggressive Solution to Rebalance the Microbiota of Acne-Prone Skin

**DOI:** 10.3390/pharmaceutics15082083

**Published:** 2023-08-03

**Authors:** Julie Leignadier, Marie Drago, Olivier Lesouhaitier, Magalie Barreau, Albert Dashi, Oliver Worsley, Joan Attia-Vigneau

**Affiliations:** 1Lucas Meyer Cosmetics, 195 Route d’Espagne, 31036 Toulouse, France; joan.attia@lucasmeyercosmetics.com; 2Shiseido EMEA, 56A Rue du Faubourg St Honoré, 75008 Paris, France; marie.drago@emea.shiseido.com; 3Research Unit Bacterial Communication and Anti-Infectious Strategies (CBSA, UR4312), University of Rouen Normandie, 27000 Evreux, France; olivier.lesouhaitier@univ-rouen.fr (O.L.); magalie.barreau@univ-rouen.fr (M.B.); 4Sequential Skin Ltd., 85 Great Portland Street, London W1W 7LT, UK; albert@sequential.bio (A.D.); oliver@sequential.bio (O.W.)

**Keywords:** acne, dendrimer, lysine, microbial diversity, clinical studies, biofilm

## Abstract

Acne is a chronic inflammatory skin disease that affects the quality of life of patients. Several treatments exist for acne, but their effectiveness tends to decrease over time due to increasing resistance to treatment and associated side effects. To circumvent these issues, a new approach has emerged that involves combating the pathogen *Cutibacterium acnes* while maintaining the homeostasis of the skin microbiome. Recently, it was shown that the use of a G2 lysine dendrigraft (G2 dendrimer) could specifically decrease the *C. acnes* phylotype (IAI) involved in acne, compared to non-acne-causing *C. acnes* (phylotype II) bacteria. In the present study, we demonstrate that the efficacy of this technology is related to its 3D structure, which, in contrast to the linear form, significantly decreases the inflammation factor (IL-8) linked to acne. In addition, our in-vitro data confirm the specific activity of the G2 dendrimer: after treatment of bacterial cultures and biofilms, the G2 dendrimer affected neither non-acneic *C. acnes* nor commensal bacteria of the skin (*Staphylococcus epidermidis*, *S. hominis,* and *Corynebacterium minutissimum*). In parallel, comparative in-vitro and in-vivo studies with traditional over-the-counter molecules showed G2’s effects on the survival of commensal bacteria and the reduction of acne outbreaks. Finally, metagenomic analysis of the cutaneous microbiota of volunteers who applied a finished cosmetic product containing the G2 dendrimer confirmed the ability of G2 to rebalance cutaneous acne microbiota dysbiosis while maintaining commensal bacteria. These results confirm the value of using this G2 dendrimer to gently prevent the appearance of acne vulgaris while respecting the cutaneous microbiota.

## 1. Introduction

Acne vulgaris is a chronic inflammatory skin disease affecting the pilosebaceous unit. Its clinical features include oily skin, noninflammatory lesions (whiteheads and blackheads), inflammatory lesions (papules and pustules), and various degrees of scarring. Acne lesions are mainly located on the face, but also on the chest, upper back, and upper arms, where the density of sebaceous glands is higher. This skin disease affects 80% of teenagers and 25% of adults over 25 years old [1]. In addition to the visible pathophysiology, acne also has several negative mental health implications for the patient—it causes discomfort, emotional stress, anxiety, and significant loss of confidence [2,3]. Several tightly interconnected main factors are involved in the development of acne. These factors include the increase in sebum production, accompanied by the hyperproliferation of lipophilic bacteria such as *Cutibacterium acnes*. These bacteria derive their energy from the production of free fatty acids via lipase-mediated degradation of sebum lipids. In turn, this accumulation of free fatty acids induces immune system activation and skin hyper-keratinization [4]. *C. acnes* is a commensal bacterium and plays a key role in acne pathophysiology. However, recent studies show various negative and positive effects of different *C. acnes* subtypes, depending on lipase activity. Indeed, *C. acnes* bacteria showing a higher lipase activity are present on acne-prone skin, whereas this activity is lower in healthy skin. Metagenomic analysis demonstrates that *C. acnes* can be divided into three phylogenetic groups: types I, II, and III, according to genome sequence and lipase activity [5,6]. In other words, these recent studies show that microbiota dysbiosis, and particularly dysbiosis of *C. acnes* phyla, plays a significant role in acne production [7,8].

The main traditional topical treatments available over the counter for acne are benzoyl peroxide and salicylic acid [3]. However, even if these treatments demonstrate positive impacts on acne skin disease, they are associated with side effects such as skin irritation, dryness, and photo-sensibility [9]. Benzoyl peroxide acts through three fundamental mechanisms: it is bactericidal towards *C. acnes*; it has mild comedolytic and anti-inflammatory properties; and it is lipophilic, concentrating inside the sebaceous follicles to produce benzoic acid and reactive oxygen species. Benzoyl peroxide exhibits direct toxicity against *C. acnes*, via inhibition of bacterial protein synthesis, nucleotide synthesis, and metabolic pathways [10]. The antimicrobial potential of benzoyl peroxide is unique, insofar as there exists no known associated bacterial resistance [11]. For its part, salicylic acid is a peeling agent and is effective in the treatment of acne due to its keratinolytic, anti-inflammatory, and bactericidal effects [12]. These two over-the-counter treatments act differently but have one common weakness: they alter the skin’s host microflora, which could open the door to negative side effects considering the known essential role of commensal skin microorganisms in preventing certain skin diseases. Microbial dysbiosis in the skin is associated, for example, with a weakening of the external barrier against pathogens [13,14,15,16]. It has been clearly demonstrated that *C. acnes* dysbiosis is associated with acne [17]. Therefore, preserving or restoring *C. acnes* diversity could be a new strategy to prevent acne and related skin disorders.

Attia-Vigneau et al. recently reported on the use of a dendrimer composed of 48 lysine residues (G2-dendrimer), an amino acid known to have antimicrobial activity with the potential to prevent acne development [18,19]. Using methods allowing the study of biofilm and membrane permeability, the authors demonstrated that the G2-dendrimer specifically perturbates the *C. acnes* involved in acne development without affecting non-acneic *C. acnes* strains [18]. This specificity is due to differences in membrane composition and polarity between acneic and non-acneic *C. acnes*, which may preferentially attract the lysine moieties to the acneic strain [20]. Importantly, G2’s 3D structure exposes the charged lysine moiety at its surface, thus enhancing attraction to acneic *C. acnes* strains. In addition, metagenomic analysis of microbiota samples from the faces of volunteers with acne-prone skin demonstrated that the application of a formula containing the G2-dendrimer re-equilibrated the skin microbiota, enabling a reduction in acne-related pathophysiology. Indeed, after the application of a cream containing G2-dendrimer, signs of acne were significantly decreased in comparison to placebo [18].

While the ability of the G2-dendrimer to decrease signs of acne and rebalance *C. acnes* dysbiosis is thus demonstrated, it remains important to show that other main commensal bacteria present on facial skin are unaffected by treatment with the G2-dendrimer to avoid the development of a new dysbiosis. Indeed, it is important to maintain balanced cutaneous bacterial communities, because these act together to control acne through their role in immune homeostasis and inflammatory responses [21]. In a healthy context, *Cutibacterium* and *Staphylococcus* strains play an essential role in the skin ecosystem, maintaining microbiota health by fighting pathogens and participating in skin homeostasis through the production of beneficial bacterial metabolites. However, these same bacteria can shift to an opportunist mode of behavior, thereby triggering skin dysbiosis associated with additional skin disorders. Indeed, thanks to the development of metagenomic analysis of skin microbiota, understanding microbial dysbiosis as observed in inflammatory skin diseases such as hidradenitis suppurativa or acne has become a strategy for restoring the homeostasis of skin microbiota and thus combating these skin diseases [22].

The objective of the present work was to evaluate how the G2-dendrimer, alone or in formulation, might decrease signs of acne by rebalancing *C. acnes* dysbiosis without affecting the balance of the wider skin microbiota population.

## 2. Materials and Methods

### 2.1. Preparation of G2-Dendrimer

As described by Attia-Vigneau et al. [18], the lysine dendrimer (G2) was produced following the protocols described in the US 2008/0206183 patent and by Collet et al. [23]. This dendrimer is grafted with 48 lysine units, and its synthesis as described in Attia-Vigneau et al. is in accordance with the 12 green chemical principles (including the use of less solvent; no use of CMR solvent; and 80% of the solvent used is water).

### 2.2. Cell Culture

Normal Human Dermal Fibroblasts (NHDF; LMC cells bank, Toulouse, France) were isolated from healthy female donors aged between 20 and 39 years old undergoing plastic surgery, with the donors’ informed consent (Alphenyx SARL., Marseille, France). NHDF cells were maintained in a complete DMEM medium containing 1% antibiotics (penicillin, streptomycin; Sigma P0781, St Louis, MO, USA) at 37 °C under 5% CO_2_ and 95% humidity.

### 2.3. IL-8 Production

NHDF cells were seeded at 2.104 cells/96-well plates in complete DMEM. After 24 h, the medium was removed, and the cells were treated with IL-1α to induce the production of IL-8 in the presence of different concentrations of either G2-dendrimer and 48 units of linear poly-l-lysin (Biosynth, FP14985, EUCODIS Bioscience GmbH, Vienna, Austria); or 10 μM dexamethasone (positive control) for an additional 24 h, alongside an untreated negative control. At the end of incubation, the release of pro-inflammatory cytokine IL-8 was quantified using an enzyme-linked immunosorbent assay (ELISA) following supplier protocol (R&D System, DY208, Bio-Techne, Minneapolis, MN, USA).

### 2.4. Bactericidal Activity

An inoculum of each of bacterial strains *C. acnes* RT5 (ATCC 11828, Fonderephar, Toulouse, France), *C. acnes* RT6 (CIP 53.117T, Fonderephar, Toulouse, France), *C. minutissimum* (CIP 100652, Fonderephar, Toulouse, France), *S. epidermidis* (CIP 81.55T, Fonderephar, Toulouse, France), and *S. hominis* (CIP 81.57T, Fonderephar, Toulouse, France) was performed in trypton salt under aerobiotic conditions or (for *C. acnes* only) anaerobiotic conditions. CFU numeration was carried out, and when 108 CFU/mL was reached, 1 mL of each inoculum was exposed to products containing 0.4% G2-dendimer, 2.5% benzoyl peroxide, 2% salicylic acid, or their respective vehicles at 20 °C for 30 min, 1 h, or 6 h in aerobiotic conditions for all strains tested except for *C. acnes*, for which anaerobiotic conditions were used. After incubation and homogenization, 1 mL of each tube was collected and completed to 10 mL with a neutralizing medium (10% polysorbate 80, 2% saponin, 2% lecithin, 0.5% natrium thiosulfate) to stop the antimicrobial activity of the products tested. Ten minutes later, ten-fold dilutions were performed on specific media: Columbia agar for *C. acnes*, and trypcase-soy agar for all other strains. After 48 h of incubation at 36 °C in anaerobiotic conditions for *C. acnes* and aerobiotic conditions for all other strains, CFU was enumerated.

### 2.5. Clinical Study with G2-Dendrimer Versus Benchmark

A total of 16 acne-prone Caucasian male and female volunteers aged 14 to 40 years old were recruited to evaluate a cream containing 1% of G2-dendrimer versus a benchmark containing 10% benzoyl peroxide (Table 1). The volunteers applied the products on clean and dry skin under normal conditions of use twice a day (in the morning and in the evening) on split face by slight massage for 28 days. On day 0 and at the end of the application, the dermatologist counted blackheads and whiteheads (non-inflammatory lesions) as well as papules and pustules (inflammatory lesions) on each hemi-face (except the nasal pyramid, the vermillion border, and the crease of the chin and the rim of the scalp).

### 2.6. Clinical Study with G2-Dendrimer in Full Formulation

A total of 22 acne-prone participants were recruited for this study, aged 22–63. The study was performed over 18 days. The skin microbiome of each participant was sampled at four time points: baseline (before wash-out), on day 4 (i.e., following 4 days of application of 100% squalene), day 10 (i.e., following 6 days of use of 0.4% G2-dendrimer in the full formulation as described below), and day 18 (i.e., following 8 days of 0.4% G2-dendrimer in formulation).

Full formulation INCI name: aqua (water), glycerin, propanediol, lactic acid, inulin, alpha-glucan oligosaccharide, *acetum* (vinegar), maris aqua (sea water), arginine, fructose, glucose, *hibiscus sabdariffa* flower extract, *laminaria digitata* extract, *chlorella vulgaris* extract, polylysine, cellulose gum, caprylyl glycol, xanthan gum, cellulose, phenethyl alcohol, saccharide isomerate, tetrasodium glutamate diacetate, sodium hydroxide, sodium benzoate, potassium sorbate.

### 2.7. Biofilm Formation on Glass Slides

*S. aureus* and *S. epidermidis* biofilm formation during exposure to G2 was evaluated on 24-well culture flat glass bottom plates (Sensoplate, Greiner bio-one, Frickenhausen, Germany). After overnight incubation (37 °C), bacteria were collected by centrifugation (7500× *g*, 10 min, 20 °C). The pellet was re-suspended in 5 mL of sterile physiologic water (SPW). Finally, 300 μL of bacterial inoculum (OD 580 nm = 0.01) was layered in the wells of 24-well culture plates. These plates were incubated for 2 h to allow primary bacterial adhesion. Then, water was removed and 1 mL of BHI medium, supplemented with the G2-dendrimer (alongside an untreated control), was added. Plates were incubated at 37 °C.

After 24 h of incubation, required for mature biofilm formation, the medium was carefully aspirated to remove planktonic cells. Next, 300 μL of SYTO 9 green-fluorescent nucleic acid stain (Thermofisher, Waltham, MA, USA) was added in each well and incubated for 20 min in the dark. The SYTO 9 stain was removed and 300 μL SPW was added in each well. CLSM observations were immediately performed using a Zeiss LSM710 (Carl Zeiss Microscopy, Oberkochen, Germany) using a 63-x oil immersion objective. SYTO 9 was excited at 488 nm and fluorescence emission was detected between 500 and 550 nm. Images were taken every micrometer throughout the whole biofilm depth. For visualization and processing of three-dimensional (3D) images, the Zen 2.1 SP1 software (Carl Zeiss Microscopy, Oberkochen, Germany) was used. Quantitative analyses of image stacks were performed using the COMSTAT software (http://www.imageanalysis.dk/) [24]. Maximal and average biofilm thickness (μm) and biomass volume (μm^3^/μm^2^) were determined. Each study was repeated a minimum of three times.

### 2.8. Skin Microbiome Sampling

Skin microbiome samples were collected using the Sequential Skin Kit, following the manufacturer’s instructions (Sequential Skin Ltd., London, UK). A Sequential Skin adhesive patch was applied on the participant’s cheek for 10 s. The patch was peeled and stored in a 15 mL tube containing a preservation solution (Sequential Skin Ltd., London, UK). Subsequently, microbial DNA was extracted following the Sequential Skin DNA extraction protocol (Sequential Skin Ltd., London, UK). Bacterial detection was performed using the Sequential Smart Probe^TM^ panel; 20 key specific skin bacterial markers were used to quantify bacterial communities using qPCR. After total DNA was collected, we assessed the purity and the quality of the samples and performed high-throughput qPCR. Raw qPCR data were cleaned, and bacterial analysis was performed. Taxonomy distribution stacked bar plots of 20 key skin microbial species across different time points were generated based on relative abundance. Absolute abundance was also calculated for 20 bacteria, including *S. epidermidis*, *S. aureus*, and *C. acnes* strain subtypes I and II, using Sequential Smart Probe^TM^ positive controls (Sequential Skin Ltd., London, UK). To further investigate global bacterial diversity changes, we assessed the diversity. The Shannon–Simpson indices were calculated to account for the richness and evenness of communities (the higher the index values, the more diverse the skin microbiome).

### 2.9. Statistical Tests

For the data obtained from the in-vitro studies, a one-way ANOVA test followed by the Bonferroni test was used. For the clinical studies, the one-way ANOVA test followed by Fisher’s LSD test was used. The statistical significance threshold was set at *p* < 0.05 (* *p* < 0.05, ** *p* < 0.01, *** *p* < 0.001).

## 3. Results

### 3.1. Anti-Inflammatory Activity of Dendrimer

The 3D structure of poly-l-lysine allows for a better distribution of lysine charges, as well as for their better accessibility. The benefit of using poly-l-lysine thus arrayed in a 3D structure, as compared to a linear structure, was confirmed by the evaluation of anti-inflammatory activity induced by IL-1α on a normal human fibroblast cell culture. The amount of pro-inflammatory cytokine IL-8 secreted after treatment with G2-dendrimer was compared to that released after treatment with linear poly-l-lysine used in equivalent unit numbers (48 units). Under these conditions, IL-8 production was significantly decreased with G2-dendrimer treatment at all concentrations tested, compared to the untreated IL-1α stimulation condition (−24% at 10^−6^ M, −27% at 10^−7^ M, −36% at 10^−8^ M, and −36% at 10^−9^ M). Conversely, IL-1α-stimulated fibroblasts showed increased IL-8 secretion in a dose-dependent manner respective to the concentration of linear poly-l-lysine (Figure 1) added in the medium (+34% at 10^−6^ M, +24% at 10^−7^ M, +19% at 10^−8^ M, and +7% at 10^−9^ M compared to IL-1α condition).

These results highlight the advantages of using poly-l-lysine arrayed into a 3D structure to produce anti-inflammatory activity.

### 3.2. Effect on Staphyloccocus Aureus and Staphyloccocus Epidermidis Biofilm Formation

In order to evaluate if the G2-dendrimer acts specifically against *C. acnes* biofilms [18], we evaluated the impact of the G2-dendrimer on the biofilm formation of *S. aureus* and *S. epidermidis*, two main bacteria of the skin microbiome [25]. Two skin commensal strains (*S. aureus* MFP03 and *S. epidermidis* MFP04) obtained from healthy skin volunteers were selected for this purpose [26,27]. We observed that both *S. aureus* and *S. epidermidis* strain exposure to G2 did not modify the ability of the two bacteria to build a biofilm. We note that even if the *S. epidermidis* biofilms exposed to G2 present a slightly different overall organization (Figure 2B), the measurement of the biovolume of these different biofilms shows no significant difference, suggesting that there is no impact on the bacterial biomass making up these biofilms. Together with our previously reported data, this suggests that G2 specifically affects the ability of the acneic *C. acnes* strain RT5 to form a biofilm, without affecting non-acneic *C. acnes* strains [18] or staphylococci. This specific effect hypothesis appears reinforced by the fact that G2 does not alter the ability of skin staphylococci to form a biofilm (Figure 2).

The microbiota is considered to be the first barrier against external aggressions, and it is essential to preserve it in order to avoid the appearance of skin disorders. With that need in mind, we evaluated ingredients with a demonstrated ability to fight the signs of acne for their ability to also maintain the survival of commensal bacteria on the skin. G2-dendrimer, shown to be capable of fighting acne by rebalancing *C. acnes* strains [18], was compared with over-the-counter ingredients known to reduce the signs of acne, salicylic acid, and benzoyl peroxide [13]. Several of the main commensal bacteria species found on the skin surface, such as *C. acnes* (acneic and non-acneic strains), *S. epidermidis*, *S. hominis*, and *C. minutissimum*, were incubated with the above active ingredients, and the evolution of bacteria survival over time was evaluated. As shown in Figure 3, salicylic acid had a drastic effect, reducing all commensal bacteria tested by 50% after an interaction of less than 20 min. The effect of benzoyl peroxide on bacteria survival was less immediate, and more dependent on the bacterial strain studied—reducing bacteria survival by 50% after approximatively 1 h for non-acneic *C. acnes* and *C. minutissimum* (Figure 3B,D), after 2.5 h for *S. hominis* (Figure 3E), and after over 3.5 h for acneic *C. acnes* and *S. epidermidis* (Figure 3A,C). In contrast, G2-dendrimer had no bactericidal activity on any of the commensal bacteria studied, and the bacterial populations were shown to be maintained over time (Figure 3). The vehicles used for each condition have no bactericidal activity on commensal bacteria studied (personal observation, 2021). These results demonstrate that the G2-dendrimer of lysine has a gentle action on skin microbiota.

### 3.3. Efficacy of G2-Dendrimer Versus Benzoyl Peroxide in a Clinical Study

The application of a cream containing G2-dendrimer has been shown to rebalance *C. acnes* strains in favor of non-acneic bacteria, potentially helping reduce signs of acne [18]. Here, we have demonstrated in addition that G2-dendrimer better maintains the microbiome compared to two commonly used over-the-counter molecules (Figure 3). The next step was to demonstrate the potential of this microbiota modulation as a solution against acne in a clinical study comparing the G2-dendrimer with benzoyl peroxide. For that purpose, acne-prone volunteers applied a cream containing G2-dendrimer or a benzoyl peroxide benchmark cream for 28 days, following a split-face protocol. The number of non-inflammatory lesions and pustules was evaluated on day 0 and day 28 by a dermatologist. Matching previous data [18], we confirmed that G2-dendrimer application significantly decreased (−21%) the number of non-inflammatory lesions compared to day 0 (Figure 4A). Moreover, the effect of G2-dendrimer on the number of noninflammatory lesions was stronger than that of the reference product, for which no statistically significant changes were observed between day 0 and day 28 (Figure 4A). The same effects were observed even more clearly in the pustule counts, where the G2 dendrimer effected a strong decrease (72%), while no change was observed with the benchmark (Figure 4B).

Taken together, these results demonstrate that the gentle action of G2-dendrimer can effectively reduce the signs of acne.

### 3.4. Impact of G2-Dendrimer on Commensal Bacteria

In the study discussed above, we showed that the application of G2-dendrimer in a simple formulation (Table 1) significantly decreases the signs of acne on volunteers with acne-prone skin, possibly due to a demonstrated rebalance of *C. acnes* strains in favor of non-acneic bacteria [18]. Next, we evaluated *C. acnes* strain distribution on volunteers who applied G2-dendrimer formulated in a more complex formulation. For that purpose, we evaluated several diversity indices taking into account both species richness (number of different species in a community) and evenness (relative abundance of species) on volunteers who applied a formulation containing the G2-dendrimer [28]. In this study, microbiota diversity was quantified using the Shannon index, a commonly recognized index of richness, as well as the Simpson index, which is more sensitive to species evenness [29]. As shown in Figure 5A,B, the Shannon and Simpson indexes remained unchanged after day 10 and day 18 of product usage application. This suggests that the formulation containing G2-dendrimer generally preserved microbiome diversity and did not affect the richness and diversity of commensal bacteria. In parallel, we performed a quantitative taxonomic analysis of 20 key skin bacterial species across 4 different time points over the 18-day study. As shown in Figure 5C, the relative abundance of total *C. acnes* species was increased at day 18 when compared with baseline, day 4, and day 10, suggesting a strong re-growth of this bacteria after two weeks of G2-dendrimer-based product usage. This was confirmed by using a targeted qPCR approach, where we quantified the total *C. acnes* genome and showed that total *C. acnes* strains significantly increased after application of the G2-dendrimer-based product (Figure 5D).

Taken together, these results indicate that the G2-dendrimer modulates the *C. acnes* family without adversely affecting the diversity and evenness of other commensal bacteria.

### 3.5. Distribution of C. acnes

Sorel Fitz Gibbon’s group has demonstrated that the relative quantity of *C. acnes* in acneic skin and healthy (acne-free) skin is similar, suggesting that it is the distribution between *C. acnes* strains that might be responsible for an acneic phenotype [6,29]. Following up on the distributions shown in Figure 5, we investigated the variation in the populations of acneic and non-acneic *C. acnes*, as well as *S. epidermidis* and *S. aureus*, by qPCR on volunteers who applied G2-dendrimer in full formulation, as compared to the absolute abundance of these bacteria at baseline (Figure 6). The results showed that the application of a complex formula containing the G2-dendrimer tended to decrease the absolute abundance of acneic *C. acnes* (Figure 6A), whereas, in contrast, non-acneic *C. acnes* tended to increase (Figure 6B). Moreover, as previously observed, the G2-dendrimer had no impact on the absolute abundance of *S. epidermidis* and *S. aureus*, with no significant difference between day 0 and 18 days of application of the full formulation containing G2-dendrimer. (Figure 6C,D).

## 4. Discussion

The aim of this work was to investigate the potential efficacy of G2-dendrimer treatment in managing the dysbiosis associated with acne, rebalancing the microbiota toward a healthier equilibrium. Acne is a chronic inflammatory skin disease commonly affecting adolescent and adult populations. Several over-the-counter treatments exist, but they are associated with undesirable side effects and may adversely affect the overall skin microbiota balance [9,13,14,15,16].

The skin microbiota is composed of millions of bacteria, fungi, and viruses, and plays essential roles in protecting the skin and the organism as a whole from foreign pathogens, conditioning the immune system, and metabolizing natural products [21,30,31]. With the rise of the microbiome research field, new findings have led to a better understanding of the relationship between skin microorganisms and acne [6,32,33]. While *C. acnes* is commonly found in sebum-rich areas and its over-proliferation has long been thought to contribute to acne pathology, it is also considered an important commensal bacterium for skin health. Indeed, *C. acnes* prevents colonization and invasion by pathogens via the hydrolysis of triglycerides in sebum and the release of antimicrobial fatty acids that contribute to the maintenance of an acidic pH at the skin surface [34].

During an acne episode, a local environment shift leads to the selection of distinct strains genetically capable of adapting to their new environment. Indeed, Fitz-Gibbon’s group demonstrates by metagenomic analysis that the subgroup of *C. acnes* with reduced lipase activity (non-acneic *C. acnes*) has a growth disadvantage in a multiphyletic community competing for space and resources compared to the subgroup of *C. acnes* with high lipase activity (acneic *C. acnes*) [6]. This *C. acnes* dysbiosis in acne-affected skin leads to an increase in the relative abundance of *S. epidermidis* [35,36]. Specific *Staphylococcal* species on the skin can secrete lantibiotics with potent antimicrobial activity against *S. aureus* [37]. These studies underline the fact that dysbiosis of one bacterial species can alter the entire skin microbiota and promote the development of skin diseases. This close, balanced relationship between different strains of bacteria is important for maintaining microbiota homeostasis for skin health.

A new technology developed by Attia’s team utilized the antibacterial power of the amino acid lysine and the 3D structure of dendrimers to specifically disrupt the membrane of acneic *C. acnes* under acneogenic conditions [18]. Ɛ-poly-l-lysine moieties built into the G2-dendrimer bind onto the bacterial cell surface, playing an important role in bactericidal activity [23,38]. The surface presentation of poly-l-lysine in a 3D structure obtained through dendrimer technology increases this adsorption to the membrane of bacteria and thus enhances the disruptive activity of lysine [18]. In addition, poly-l-lysine is known to have anti-inflammatory activity at high concentrations [39]; the anti-inflammatory effect of poly-l-lysine also seems enhanced when presented in a 3D dendrimer structure. As shown in Figure 1, the efficacy of the lysine dendrimer is higher than that of linear poly-l-lysine (the concentration required for efficacy with the dendrimer is lower than that needed with a linear lysine polymer). The low concentration of use of the G2-dendrimer may enable better specificity and targeting of bacteria possessing the most attractive charge for lysine. Indeed, the composition and surface polarity of acneic and non-acneic strains of *C. acnes* are different, resulting in a specificity of action of the G2-dendrimer depending on the membrane properties of the respective bacterial strains [18,20].

This G2-dendrimer is a new technology that enables a more specific application of the bactericidal activity of lysine (Figure 3), allowing for use at lower concentrations and targeting acneic *C. acnes*. The effectiveness of treatment using a poly-l-lysine dendrimer is highlighted by the observed decrease in the signs of acne on the face of volunteers with acne-prone skin compared to traditional treatments that are currently available over the counter, and which require strong bactericidal activity to be effective. Benzyl peroxide, used as a benchmark in this study, has comedolytic action and the ability to kill *C. acnes* via the release of free oxygen radicals that may disrupt the equilibrium of the skin microbiota [39]. Finally, a treatment that rebalances the microbiota is more effective in removing the signs of acne (Figure 4) than benzyl peroxide, which negatively impacts the microbiota. Using the G2-dendrimer in a complex formula does not affect the efficacy of the G2-dendrimer, and we observed a rebalancing of the skin microbiota in favor of non-acneic *C. acnes* strains, without significant changes in other main skin bacteria.

## 5. Conclusions

The G2-dendrimer discussed here is a new technology that conserves the overall skin microbiome balance whilst affecting the *C. acnes* family in a strain-specific manner, thereby significantly improving acne pathophysiology.

## 6. Potential Limitation

The limitation of our study is that the degree of acne varied between volunteers, even though a minimum number of lesions was required when setting up the clinical study. Moreover, the microbiome also varied between volunteers and between different communities. These two criteria limit the generalizability of the results. Furthermore, as the study was conducted over 28 days, it will be interesting to observe whether the restoration of cutaneous dysbiosis is maintained over time to avoid the reappearance of skin disorders.

## Figures and Tables

**Figure 1 pharmaceutics-15-02083-f001:**
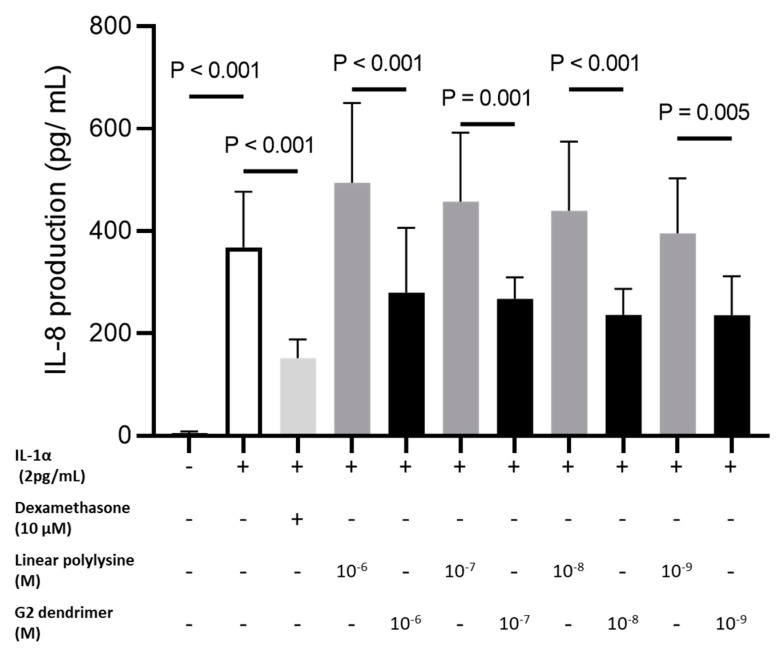
Anti-inflammatory effect of dendrimer versus linear poly-L-lysin. NHDF pre-stimulated with IL-1α to induce inflammation were treated with dexamethasone (DXM) at 10 μM as a positive control, or with G2-dendrimer versus 48 units of linear polylysine at different concentrations (10^−6^ M to 10^−9^ M). The production of IL-8 was normalized to cell viability (MTS test). Data are presented as the mean of IL-8 + standard deviation.

**Figure 2 pharmaceutics-15-02083-f002:**
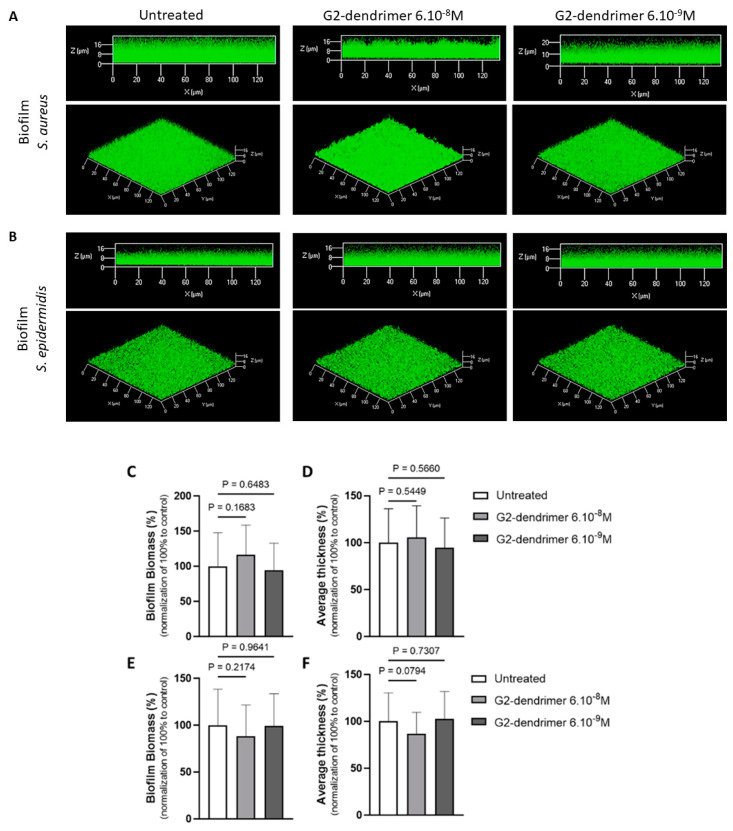
No impact of the G2 dendrimer on staphylococcus aureus and *S. epidermidis* biofilm. The 3D shadow representations of the biofilms (side view (upper part) and biofilms structure (lower part)) developed by *S. aureus* MFP03 (**A**) and *S. epidermidis* MFP04 (**B**) strains exposed to G2 dendrimer (6 × 10^−8^ M or 6 × 10^−9^ M) vs. untreated control. COMSTAT analyses of biofilms biomasses (**C**,**E**) and of the average thickness (D/F) concerning biofilms of S. aureus MFP03 (**C**,**D**) and of *S. epidermidis* MFP04 (**E**,**F**) strains exposed to G2 dendrimer (6 × 10^−8^ M or 6 × 10^−9^ M) vs. untreated control. Data are presented as the means of at least 27 measurements from three independent experiments. NS: no statistically significant difference.

**Figure 3 pharmaceutics-15-02083-f003:**
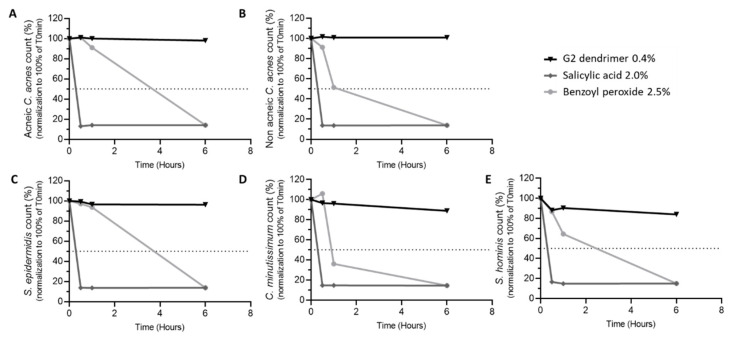
Anti-bactericidal effect of dendrimer G2. (**A**) acneic *C. acnes*, (**B**) non acneic *C. acnes*, (**C**) *S. epidermidis*, (**D**) *C. minutissimum*, and (**E**) *S. hominis* bacterial strains were inoculated with dendrimer G2 at 0.4%, salicylic acid at 2%, or benzoyl peroxide at 2.5%. Bacterial populations were enumerated 0.5, 1, and 6 h later. Data are presented as counts ± standard deviation, normalized to 100% of T0.

**Figure 4 pharmaceutics-15-02083-f004:**
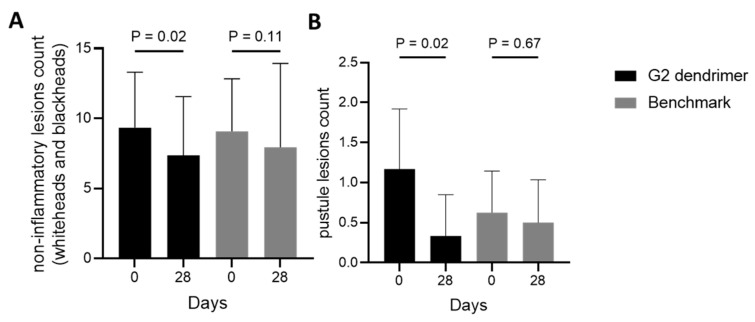
Comparison of G2-dendrimer vs. benchmark on signs of acne: non-inflammatory lesions (**A**) and pustules (**B**) were enumerated twice a day, for 28 days, on the face of volunteers before and after the application of a cream containing G2-dendrimer (black) versus a benzoyl peroxide benchmark (gray) Results are presented as mean ± SD.

**Figure 5 pharmaceutics-15-02083-f005:**
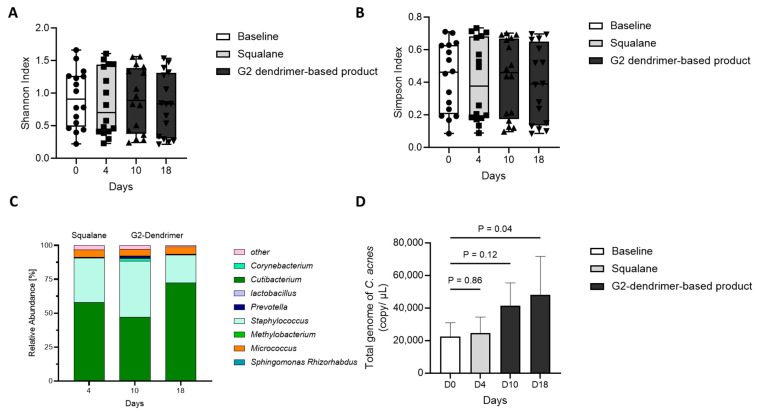
Conserved bacterial diversity after application of G2-dendrimer in full formulation. (**A**,**B**) Box plots showing Shannon diversity (**A**) and Simpson diversity index (**B**), distribution of numerical data, and skewness, showing the first (lower) quartile, median, and third (upper) quartile. (**C**) Stacked bar plots showing relative abundance across 20 key bacterial species grouped into species. Each color represents a bacterial genus, and the height of the color block indicates the relative abundance of that species before and after the application of cream containing G2-dendrimer. (**D**) Total *C. acnes* genome by qPCR, as the mean of absolute abundance (genome copy/µL) ± SEM.

**Figure 6 pharmaceutics-15-02083-f006:**
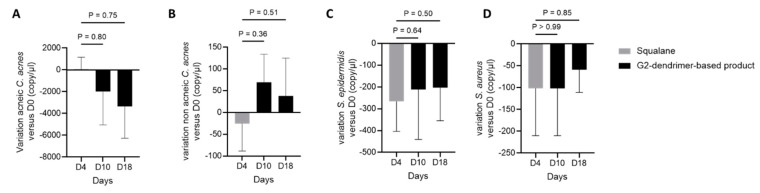
Bacterial populations after application of G2-dendrimer production full formulation. The genomes of acneic (**A**) and non acneic (**B**) *C. acnes*, *S. epidermidis* (**C**), and *S. aureus* (**D**) were determined by qPCR. Bar graphs represent the mean of the variation of absolute abundance (genome copy/µL) vs. baseline absolute abundance + SEM.

**Table 1 pharmaceutics-15-02083-t001:** Clinical trial formulation.

Ingredients (Trade Name)	INCI Name	Active Lotion (%)
Dermofeel PA-3	Sodium phytate (and) aqua (and) alcohol	0.10
Ecogel	Lysolecithin (and) sclerotium gum (and) xanthan gum (and) pullulan	2.00
SCB jojoba oil	Simmondsia chinensis seed oil	3.00
Dermofeel toco 70 non gmo	Tocopherol (and) helianthus annuus (sunflower) seed oil	0.10
Lipex shea W	Shea butter cetyl esters	3.00
Saboderm TCC	Caprylic/capric triglyceride	3.00
Dekaben C4	Phenoxyethanol (and) methylparaben (and) ethylparaben (and) butylparaben (and) propylparaben	0.80
G2-dendrimer	Glycerin (and) water (and) polylysine	0.0005
TOTAL		100.00

## Data Availability

The data presented in this study are available on request from the corresponding author.

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
