# Peer review of "Lysine-Dendrimer, a New Non-Aggressive Solution to Rebalance the Microbiota of Acne-Prone Skin"

_pharmaceutics, 2023, doi:10.3390/pharmaceutics15082083_

Round 1

Reviewer 1 Report

The article is very interesting and well written

The material and methods section explains the methods used very well, the figures are of a good standard. I would have just minor revisions for the authors

1) Add a paragraph where they talk about the limitations of the study in order to allow for more discussion in the future

2) In the introduction we need to discuss the role of the microbiota in general, its importance is of great importance these years not only in acne but also in other related diseases such as HS. I leave a reference that the authors can use

- DOI: 10.1111/ced.15291

3) The discussion and conclusion section should be divided.

4) Minimal language revision is required

Minor editing of English language required

Reviewer 2 Report

Great work! Please provide some minor changes in methodology part in order to explain first 3 comments. I am interested to read the answer to 4th comment.

Comment 1: Why in clinical study with G2-dendrimers for the control placebo cream (without G2-dendrimers) was not used?

Comment 2: Explain methodology „dermatologist counted non-inflamatory lesions“ in more detail.

Comment 3: Explain why 100% Squalene was used in first 4 days in clinical study with G2-dendrimers in full formulation.

Comment 4: Do you have any data for any clinical studies with placebo formulation?
